# Diatom Diversity on the Skin of Frozen Historic Loggerhead Sea Turtle Specimens

**Lucija Kanjer [1], Roksana Majewska [2,3], Bart Van de Vijver [4,5], Romana Gračan [1], Bojan Lazar [6,7] and Sunčica Bosak [1,*]**

[1] Department of Biology, Faculty of Science, University of Zagreb, Rooseveltov trg 6, 10000 Zagreb, Croatia; lucija.kanjer@biol.pmf.hr (L.K.); romana.gracan@biol.pmf.hr (R.G.)
[2] Unit for Environmental Sciences and Management, School of Biological Sciences, North-West University, Private Bag X6001, Potchefstroom 2520, South Africa; roksana.majewska@nwu.ac.za
[3] South African Institute for Aquatic Biodiversity (SAIAB), Private Bag 1015, Grahamstown 6140, South Africa
[4] Research Department, Meise Botanic Garden, Nieuwelaan 38, B–1860 Meise, Belgium; bart.vandevijver@plantentuinmeise.be
[5] Department of Biology, ECOBE, University of Antwerp, Universiteitsplein 1, B–2610 Antwerpen, Belgium
[6] Department of Biodiversity, Faculty of Mathematics, Natural Sciences and Information Technologies, University of Primorska, Glagoljaška 8, 6000 Koper, Slovenia; bojan.lazar@upr.si
[7] Marine Sciences Program, University of Pula, Zagrebačka ul. 30, 52100 Pula, Croatia
* Correspondence: suncica.bosak@biol.pmf.hr

**Abstract:** In recent years, biofilm-forming diatoms have received increased attention as sea turtle epibionts. However, most of the research has focused on carapace-associated taxa and communities, while less is known about diatoms growing on sea turtle skin. The current study investigated diatom diversity on the skin of loggerhead sea turtle heads detached from the carcasses found along the Adriatic coast between 1995 and 2004 and stored frozen for a prolonged period of time. By using both light and scanning electron microscopy we have found diatom frustules in 7 out of 14 analysed sea turtle samples. Altogether, 113 diatom taxa were recorded, with a minimum of seven and a maximum of 35 taxa per sample. Eight taxa, *Achnanthes elongata*, *Berkeleya* cf. *fennica*, *Chelonicola* sp., *Licmophora hyalina*, *Nagumoea* sp., *Navicula* sp., *Nitzschia* cf. *lanceolata*, and *Poulinea lepidochelicola* exceeded 5% of relative abundance in any one sample. The presumably obligately epizoic diatom taxa, *A. elongata*, *Chelonicola* sp., and *P. lepidochelicola*, dominated in six loggerhead samples, contributing up to 97.1% of the total diatom abundance. These observations suggest that on the sea turtle skin highly specialised taxa gain even greater ecological advantage and dominance over the co-occurring benthic forms than in the carapace biofilms. The suitability of frozen sea turtle skin specimens for diatom analysis and limitations of this approach are discussed.

**Keywords:** epibiont; *Bacillariophyta*; *Caretta caretta*; Mediterranean Sea; marine gomphonemoid diatoms; epizoic biofilm

## 1. Introduction

With the recent increased interest in diatom diversity on marine vertebrates [1–4] in general and sea turtles in particular [5–10], a considerable number of sea turtle-associated diatom taxa have been described within the last five years [11–24] and it became clear that diatoms constitute an important element of the sea turtle epi-microbiome on both juvenile and adult individuals. Several sea turtle-associated diatom species have thus far only been reported from this substratum, suggesting a very close, even possibly obligatory relationship between the sea turtle-associated diatoms and their hosts. However, to date, more attention has been given to diatoms growing on sea turtle carapaces,

although there is growing evidence that these may differ significantly from the skin-associated taxa and assemblages [10–12,16,24].

Except for leatherbacks (*Dermochelys coriacea* Vandelli, 1761), sea turtles possess bony carapaces covered with keratinous plates or scutes, while their skin, being both durable and flexible, is only moderately keratinised [25]. As in other reptiles, the external layer of the sea turtle skin contains both α-keratin and β-keratin concentrated mainly in the continuously shedding scales [26]. The shedding patterns of the sea turtle skin scales differ from those typical of the hard shells, in which scutes are either retained or shed periodically [26]. Therefore, it is not surprising that sea turtle carapaces are usually more heavily colonised with epibiotic diatoms than their skin [10]. Immersed surfaces undergo a generally similar sequence of events leading to the development of the well-established biofilm [27]. The initial biochemical conditioning of substrates (including absorption of dissolved macromolecules) is followed by bacterial colonisation and subsequently the attachment of unicellular eukaryotes such as diatoms, other protists and multicellular organisms (e.g. seaweeds, small invertebrates). In sea turtle epibiosis, these initiatory processes involving microorganisms are far less explored than macroepibiosis [28], and it is possible that, for example, different types of microornamentation present on coarser carapace scutes and fine skin scales may affect the attachment ability of various pioneering, surface-conditioning microbes [27]. Similarly, although information about the potential influence of the sea turtle skin glands and their products on the skin-associated microorganisms is lacking, it is conceivable that this physiologically active substratum will provide highly specific life conditions for the biofilm developing on its surface.

Historic specimens of sea turtles may constitute an excellent source of epizoic diatom frustules, and conserved carapaces of both freshwater [29] and marine turtles [7,16,30], even those collected several decades ago [7], can be used for a diatom analysis of communities associated with animal samples and carcasses. Zoological specimens, therefore, could provide information about not only epizoic diatom diversity but also any spatial and temporal changes in the diatom community composition. However, given the perishable nature of the animal soft tissues, it was unclear whether sea turtle skin material would be similarly suitable for the epizoic diatom surveys. The current study aimed to describe the diatom community composition on historic specimens of loggerhead sea turtle (*Caretta caretta* L. 1758) heads and assess the utility of frozen sea turtle skin specimens in the exploration of epizoic diatom diversity.

## 2. Materials and Methods

### 2.1. Sampling

The samples of loggerhead skin and beak scrapings were collected on 15 April 2017 from 14 frozen loggerhead turtle heads (Table 1). All turtles were found washed up dead on several beaches along the eastern coast of the Adriatic Sea from 1995 to 2004 and taken for post-mortem examination at the Natural History Museum, Zagreb, Croatia. Animal sampling was carried out under the permit Nos. 612-07/97-31/67 and 531-06/1-02-2 of the Ministry of Environmental Protection and Physical Planning of Croatia and the permits Nos. 354-09-66/00 and 35714-165/01 of the Ministry of the Environment, Spatial Planning and Energy of Slovenia. After the necropsies, heads were frozen and stored at −20 °C for subsequent craniometrical analysis. Upon the collection of diatom samples, the heads were defrosted and multiple skin samples of approx. 20 cm$^2$ in total were collected from the head and upper part of the neck area using sterile scalpels and tweezers. In addition, scrapings from the upper part of the beak were collected using sterile scalpels (Figure 1). Each sample represents all material collected from a single turtle with corresponding sample code (Table 1). Samples were preserved in 4% formaldehyde and stored at 4 °C until further processing.

**Table 1.** Turtle sample code, locality and date of the carcass finding, cause of death, carapace size (SCCL—standard curved carapace length, CCW—curved carapace width) and sex of the sampled loggerheads.

| Sample Code | Locality | Date | Cause of Death | SCCL (cm) | CCW (cm) | Sex |
|---|---|---|---|---|---|---|
| HPM3 | Mali Lošinj, (Croatia) | 10 May 2002 | stranded | 60.7 | 56.8 | ? |
| HPM9 * | Piran Bay (Slovenia) | 1995 | stationary net | 26.6 | 25.0 | male |
| HPM23 | Neretva, Komin (Bosnia and Herzegovina) | 21 June 2001 | stationary gill nets | 57.5 | 53.0 | female |
| HPM24 | Dugi otok (Croatia) | 4 February 2002 | floating in the sea | 58.6 | 51.5 | female |
| HPM25 * | Palagruža (Croatia) | 23 April 2002 | floating in the sea | 84.5 | 75.0 | female |
| HPM31 | Prevlaka, Konavle, (Croatia) | 20 September 2002 | longline | 41.4 | 37.1 | male |
| HPM33 * | Lokrum, (Croatia) | 15 August 2002 | gill net | 40.4 | 37.0 | female |
| HPM44 | Zabodarski Bay, Lošinj (Croatia) | 1 December 2003 | stranded in the beach | 63.0 | 58.8 | female |
| HPM48 * | Poreč (Croatia) | 19 October 2002 | stranded | 79.2 | 69.2 | female |
| HPM67 | Pula (Croatia) | 1 June 2003 | floating in the sea | 58.2 | 53.2 | male |
| HPM68 | North Adriatic (Croatia) | unknown | unknown | 47.7 | 41.8 | male |
| HPM69 * | Medulin (Croatia) | 21 May 2004 | stranded | 51.3 | 46.3 | female |
| HPM70 * | Krk (Croatia) | 2 June 2004 | unknown | 38.2 | 35.5 | female |
| HPM71 * | Mali Lošinj, (Croatia) | 19 May 2004 | unknown | 32.7 | 28.8 | male |

* samples that contained diatoms.

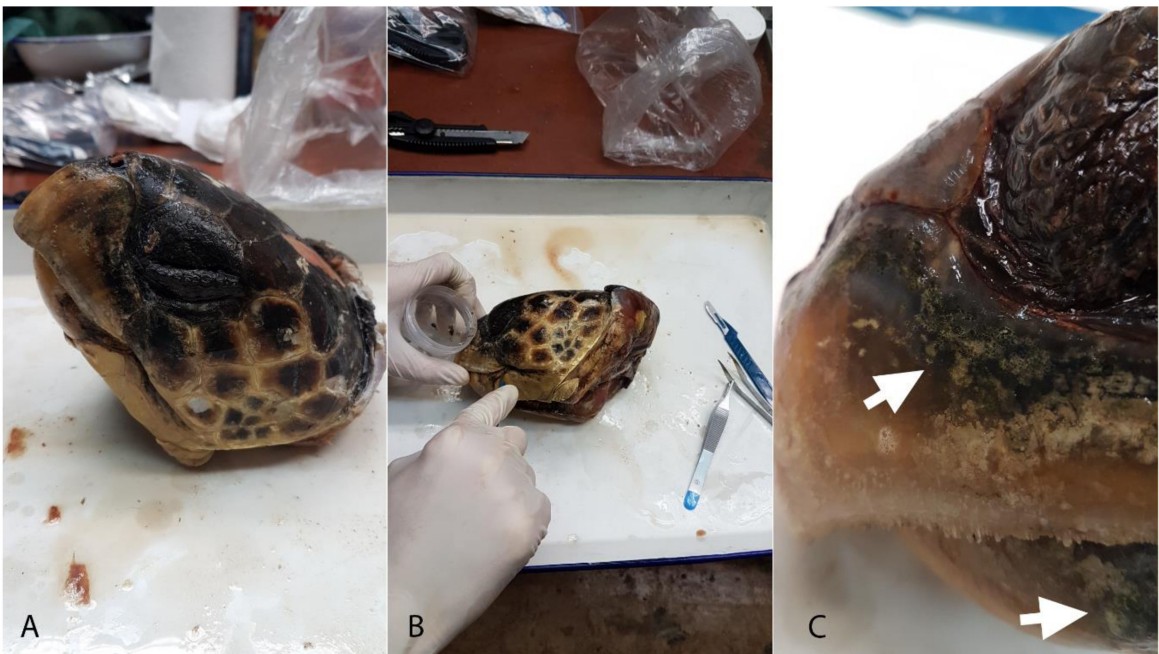

**Figure 1.** Sampling of diatoms associated with frozen loggerhead sea turtle heads; (**A**)—defrosted sea turtle head; (**B**)—biofilm scraping from a loggerhead beak; (**C**)—visible pigmented biofilm (arrows) on a loggerhead beak.

*2.2. Sample Preparation*

Samples containing pieces of skin and beak scrapings were digested with a mixture of sulphuric and nitric acids added at a ratio of 3:1 while heating on a hot plate, following the method described in [31]. The samples were washed with distilled water, centrifuged 5 times at 3700× *g*, and the resulting clean material was pipetted onto glass coverslips, dried overnight, and mounted in Naphrax (Brunel Microscopes Ltd., Wiltshire United Kingdom).

Identification and the counting of diatoms was performed using a light microscope (LM) Zeiss Axio Imager A2 equipped with DIC and ZEN Imaging software 2.5 (Carl Zeiss Microscopy GmbH, Jena, Germany). Slides were examined at 1000× magnification and at least 400 diatom valves were enumerated on random linear transects on each permanent slide. In slides HPM33, HPM71, and HPM25, only 286–325 diatom valves were counted due to low valve abundances in the original samples (Table 2). The species abundances were expressed as relative abundances (%) of the total diatom valve numbers counted in each sample.

**Table 2.** Number of diatom taxa found (S), number of valves counted (N), and values of the Margalef's species-richness (d), Pielou's evenness (J'), and Shannon–Wiener diversity (H') indices calculated for each turtle sample.

| Turtle Sample Code | S | N | d | J' | H' |
|---|---|---|---|---|---|
| HPM9 | 7 | 413 | 1.00 | 0.41 | 0.80 |
| HPM69 | 30 | 414 | 4.81 | 0.43 | 1.45 |
| HPM70 | 35 | 432 | 5.60 | 0.44 | 1.57 |
| HPM25 | 20 | 325 | 3.27 | 0.55 | 1.64 |
| HPM48 | 15 | 421 | 2.32 | 0.39 | 1.05 |
| HPM33 | 19 | 286 | 3.18 | 0.38 | 1.12 |
| HPM71 | 19 | 291 | 3.17 | 0.52 | 1.53 |

Cleaned diatom material was filtered through 3-μm Nucleopore (Nucleopore, Pleasanton, CA, USA) polycarbonate membrane filters, air-dried and mounted on aluminum stubs. Stubs were

sputter-coated with a platinum layer of 20 nm and studied using a JEOL-7100F SEM microscope (Jeol, Tokyo, Japan) at 2 kV located at Meise Botanic Garden, Meise, Belgium.

For diatom species identification selected identification books were consulted [32–34]. The nomenclature of recorded taxa follows AlgaeBase [35]. Diatoms were identified at the species level when possible, otherwise, identification was made to a genus level. Valves belonging to the *Poulinea*/*Chelonicola* complex, often impossible to distinguish using only LM, were separated based on the shape and size of the central area: valves with a clear rectangular or slightly bowtie-shaped fascia were counted as *Poulinea*, whereas those with a smaller central area as *Chelonicola*. Species identification was later confirmed using detailed observations with SEM. Permanent slides and prepared material are deposited in the Croatian National Diatom Collection (HNDC) at the Department of Biology, Faculty of Science, University of Zagreb, Zagreb (Croatia).

*2.3. Data Analyses*

Diatom abundance data were square root-transformed to downweigh dominant taxa. To avoid excessive noise in the dataset, only taxa with a relative abundance of at least 1% in one sample were included in all further statistical analyses. Two-dimensional non-metric multidimensional scaling (NMDS) based on Bray–Curtis similarity matrix was used to reveal the patterns in taxa composition between samples. Two sampling designs, one using two distinct age groups (turtles with carapace length < 50 cm as young and turtles with carapace length > 50 cm as older) and the second using years of turtle findings, were used to perform a distance-based permutational multivariate analysis of variance (PERMANOVA) [36]. The PERMANOVA pairwise test was performed on the Bray–Curtis similarity matrix of square root-transformed data, using Type III sums of squares (i.e., partial sums of squares), with fixed effects and unrestricted permutation of raw data (9999 permutations). All multivariate analyses were performed using the software packages PRIMER v6 and v7 [37], including the add-on package v6 PERMANOVA+.

## 3. Results

A total of 7 out of 14 loggerhead skin samples did not contain diatom valves and thus were excluded from further analyses (Table 1). A total of 113 diatom taxa belonging to 43 genera were found (Table S1, Figures S1 and S2), with an average of 20.7 taxa per sample. Samples HPM70 and HPM9 contained the highest (35) and the lowest number of taxa (7), respectively. Similarly, Margalef's species-richness index was highest in sample HPM70 (5.6) and lowest in sample HPM9 (1; Table 2) Pielou's evenness index reached its highest and lowest values in samples HPM25 (0.55) and HPM33 (0.38), while Shannon–Wiener diversity index in samples HPM25 (1.64) and HPM9 (0.8), respectively (Table 2).

Eight taxa, *Achnanthes elongata* (Majewska and Van de Vijver), *Berkeleya* cf. *fennica*, *Chelonicola* sp., *Licmophora hyalina* (Kützing) Grunow, *Nagumoea* sp. 1, *Navicula* sp. 1, *Nitzschia* cf. *lanceolata*, and *Poulinea lepidochelicola* Majewska et al., exceeded 5% of relative abundance in any one sample (Table S1). Only the latter occurred in all seven samples contributing up to 74.8% of the total diatom number, with *A. elongata* and *Chelonicola* sp. being present in five samples (Table S1). Together, the eight genera including the above-mentioned most abundant species contributed at least 80.1% (sample HPM70) and up to 98.8% (sample HPM9) of the total diatom number (Figure 2). Moreover, the presumably obligately epizoic diatom taxa, *A. elongata*, *Chelonicola* sp., and *P. lepidochelicola*, dominated in all but one (sample HPM70) loggerhead samples that contained diatoms (Figures 2–4) reaching a maximum of 97.1% relative abundance in sample HPM9 (Table S1). The relative abundance of 83 diatom taxa did not exceed 1% in any of the samples. These taxa, contributing altogether 0.2–8.9% of the total diatom abundance, were subsequently removed from further comparative analyses.

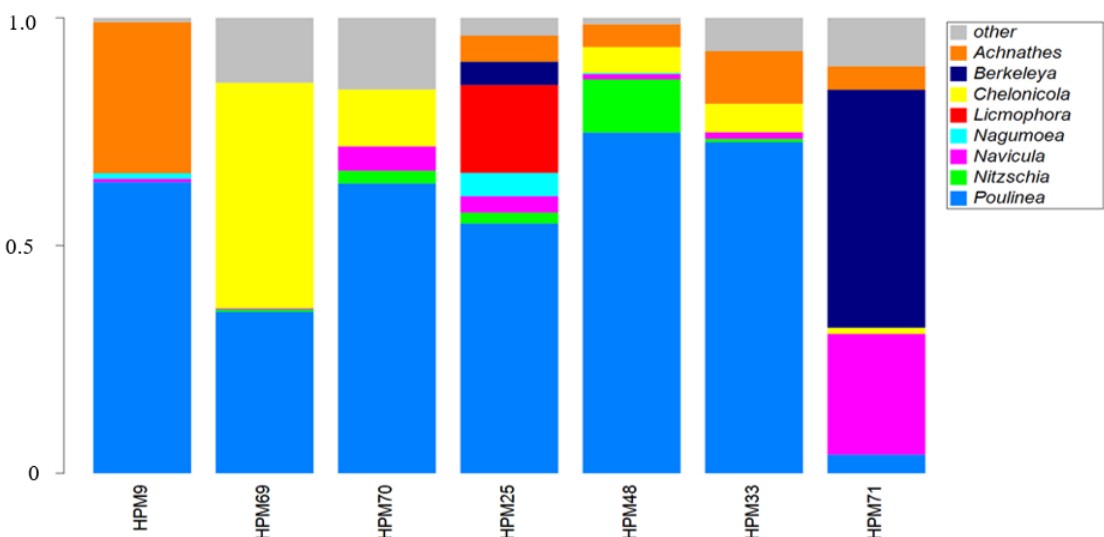

**Figure 2.** Relative abundance of eight most important genera found in examined skin samples of loggerhead sea turtles from the Adriatic Sea.

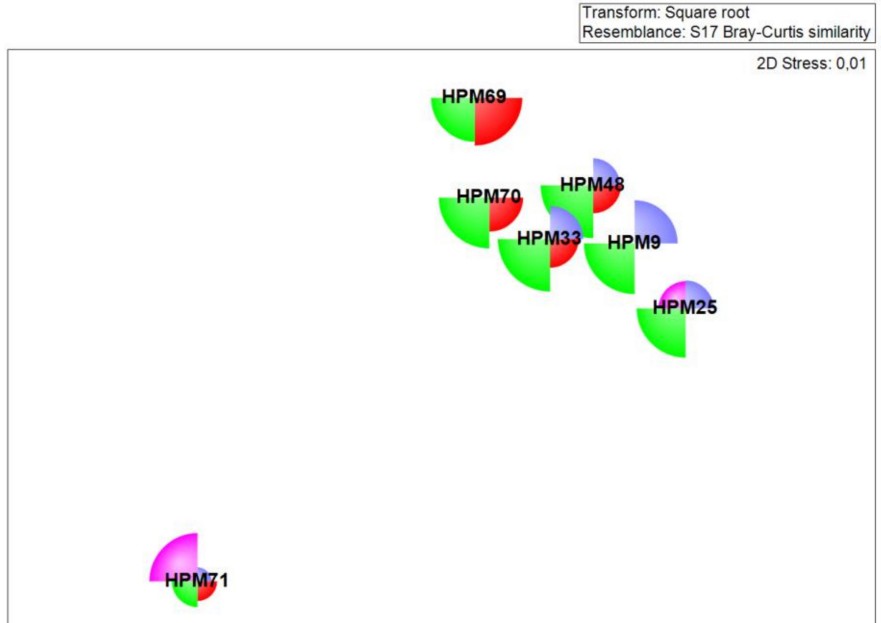

**Figure 3.** Non-metric multidimensional scaling (nMDS) plot of samples using the square root-transformed diatom abundance data, overlaid with the segmented bubble plot showing the contribution of four dominant diatom taxa in each sample: *Achnanthes elongata* (blue), *Berkeleya* cf. *fennica* (purple), *Chelonicola* sp. (red), and *Poulinea lepidochelicola* (green).

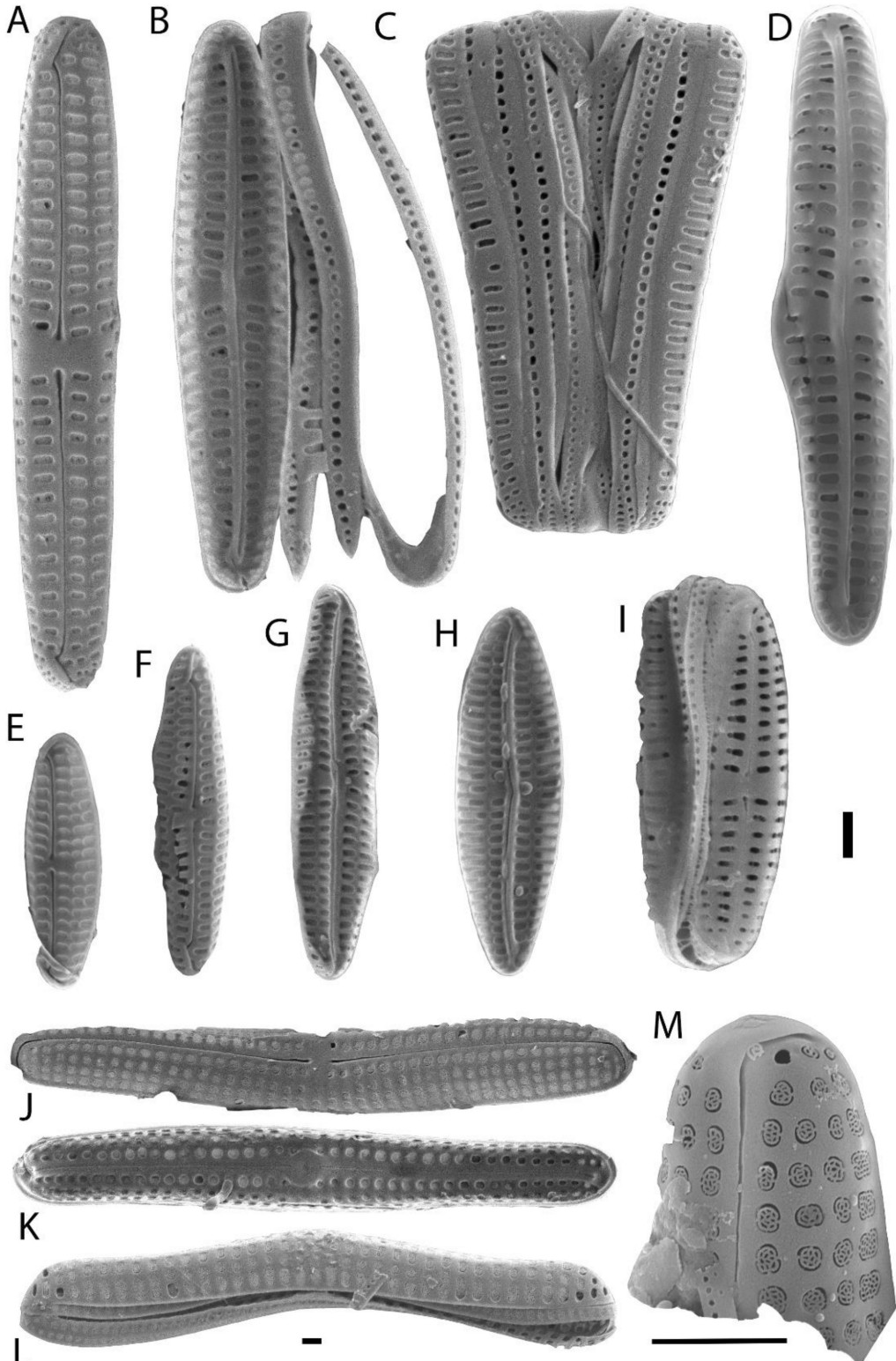

**Figure 4.** Scanning electron microscopy images of of genera *Poulinea* and *Chelonicola* specimens; (**A**) *Poulinea lepidochelicola*, outer valve view; (**B**) *P. lepidochelicola*, inner valve view and two open septate girdle bands; (**C**) *P. lepidochelicola*, girdle view of a complete frustule; (**D**) *P. lepidochelicola*, inner valve view, (**E**) and (**F**) *Chelonicola* sp., outer valve view; (**G**) and (**H**) *Chelonicola* sp., inner valve view; (**I**) *P. lepidochelicola*, outer valve view with girdle, (**J**) *Achnanthes elongata*, outer valve view (raphe valve); (**K**) *A. elongata*, inner valve view (raphe valve); (**L**) *A. elongata*, outer girdle view of a complete frustule showing the relatively deep mantle of a rapheless valve; (**M**) *A. elongata*, apical part of raphe valve showing the cribrate areolae. Scale bars represent 1 μm.

The Bray–Curtis similarity between samples calculated for standardised diatom data ranged from 38.1% (samples HPM9 and HPM69) to 86% (samples HPM33 and HPM48), except for sample HPM71 that showed from 4.6% (sample HPM9) to 10.4% (HPM25) similarity to other samples (Figure 3). The PERMANOVA indicated no significant differences between samples collected from younger and older sea turtles (Pseudo-$F = 0.801$; $p = 0.783$; $df = 6$). Similarly, no differences were detected between diatom communities collected from turtles deceased in different years (1995, 2002, and 2004; Pseudo-$F = 0.866$; $p = 0.661$; $df = 6$).

The so-called marine gomphonemoid taxa, *Poulinea* and *Chelonicola* [8,13], showed a high morphological variability. Several morphotypes were distinguished during the SEM analysis (Figure 4). The analysed specimens of *Poulinea* showed a highly variably developed apical pore field (e.g., Figure 4A vs. Figure 4I) and a distinct fascia, while valves classified as *Chelonicola* presented different sizes and shapes of their central area and areolae as well as areola number per stria, the latter ranging from 2 to 4 (Figure 4).

## 4. Discussion

The present analysis of the historic frozen head specimens of the loggerhead sea turtles showed that the skin of the Adriatic loggerheads is colonised mainly by presumably obligately epizoic taxa such as *Achnanthes elongata*, *Chelonicola* sp. and *P. lepidochelicola*, reported previously from various sea turtle species from different geographic regions [5–10,13,14,16–18,24]. Although the sample size was relatively small, the above-mentioned diatom taxa constituted an important element of the Adriatic loggerhead skin biofilm dominating in six out of seven samples containing diatoms, despite the location, cause of the host death, host age (size), or the year or season in which the sea turtle carcasses were collected. Among the 113 diatom taxa found in the sea turtle skin samples, as many as 83 contributed less than 1%. Only eight taxa contributed more than 5% of the total diatom number in any one sample. Despite the high number of species present, diversity indices calculated for the skin samples were generally lower than those obtained in the study focused on the olive ridley carapace diatoms reporting only 21 diatom taxa. These results support the hypothesis that sea turtle diatom communities are composed of the sea turtle-specific "core taxa" as well as opportunistic species that take advantage of the substratum conditioning services provided by the former group of diatoms [6] and may be associated with other micro- and macroepibionts of sea turtles that settle on the animal substratum at a later stage of the biofilm development [16]. Given the unique nature and properties of the animal skin, highly specialised taxa adapted to the epizoic lifestyle may gain even greater ecological advantage and dominance over the co-occurring benthic forms than in the carapace biofilms. Similar conclusions were inferred by Van de Vijver et al. [10], who compared diatom communities on the skin and carapace of five loggerhead individuals. In such stable communities where the pool of species able to thrive (not only survive) on the sea turtle skin under the natural conditions seems to be small, any compositional changes may indicate a functional disturbance directly or indirectly linked to the host or macrohabitat health. If so, skin diatom communities have the potential to serve as highly accurate indices of the sea turtle and ecosystem condition. For example, sample HPM71, unlike the rest of the analysed samples, was dominated by *Berkeleya* cf. *fennica*. The genus *Berkeleya* is a common biofilm-forming diatom [38] that may outcompete the specialised taxa when the growth conditions change due to essential changes in the host animal behaviour (e.g. immobilisation, free floating) or physiology (e.g. death). Future studies will likely provide more information about the causative factors involved in diatom species turnover as well as any possible seasonal or geographic variation in skin-associated communities of loggerheads and other sea turtles.

Yet again, the present study indicated that the morphological variability within the so-called marine gomphonemoid taxa, *Chelonicola* and *Poulinea*, may be extremely high even in populations collected from the same location and a single body part of one sea turtle individual. Both the intrinsic (related to these taxa biology) and extrinsic (environmental) processes are likely responsible for the observed variability and, although the explanation of these phenomena, as well as taxonomy of

*Chelonicola* and *Poulinea*, were beyond the scope of this investigation, it is worth stressing that new species within this group should be identified and described with great caution.

Half of the analysed loggerhead skin samples did not contain diatom valves. The presence or absence of diatom remnants did not seem to be related to the sea turtle age (size), sex, or the season of collection, and it is unlikely that within a sea turtle population inhabiting the same natural water body some individuals would not carry any diatoms. Therefore, it is probable that the diatom biofilm was lost due to the preservation, storage, and handling procedures of sea turtle specimens that were not intended for diatom analysis [7,29,31]. All sea turtle heads were frozen until the time of the diatom sample collection when they were thawed and sampled. However, it is unclear whether and which other preservation or cleaning protocols were applied after the sea turtle carcasses were found and how and when the heads were detached from the rest of the animal body. For example, multiple freezing and thawing cycles would disrupt the biofilm and detach the diatom frustules from their substratum. Moreover, in sea turtle carcasses collected a few days after the animal death, diatom biofilm composition might have been already altered due to the more advanced decay processes affecting the soft tissue (skin). Therefore, although the current analysis proves that frozen historic specimens containing sea turtle skin may be successfully used for epizoic diatom analysis, the above-mentioned factors possibly affecting the original diatom biofilm must be considered before drawing firm conclusions about diatom occurrences, distribution patterns, or abundances.

**Supplementary Materials:** The following are available online at http://www.mdpi.com/1424-2818/12/10/383/s1, Table S1 List of diatom taxa found in loggerhead skin samples (HPM9, HPM69, HPM70, HPM25, HPM48, HPM33 and HPM71) and their relative abundances. Figure S1 Light micrographs of diatoms found in the loggerhead skin samples. Scale bars = 5 μm. Sample HPM9: A—*Achnanthes elongata*, B—*Poulinea lepidochelicola*, C—*Nagumoea* sp., Sample HPM69: D—*Diploneis smithii*, E—*Astartiella* cf. *bahusiensis*, F—*Diploneis decipiens* var. *parallela*, G—*Chelonicola* sp., HPM69: H—*Nitzschia* sp. 7, I—*Nitzschia* sp. 8, J—*Psammodictyon* cf. *panduriforme*, K—*Amphora bigibba*, L—*Stauroneis* sp. 1, HPM33: M—*Berkeleya* cf. *fennica*, N—*Diploneis decipiens* var. *paralella*, O—*Pseudogomphonema* cf. *kamschaticum,* P—*Tursiocola* sp. 1, HPM25: R—*Cyclophora tenuis*, S—*Nitzschia* sp. 6, T—*Hyalosira* sp., U—*Cocconeis scutellum,* HPM70: V—*Cocconeis* sp. 5, W—*Nitzschia* sp. 3, X—*Fragilaria* sp. 1, Y—*Amphora* cf. *proteoides,* HPM71: Z—*Cyclotella* sp. 3, AA—*Cocconeis convexa*, AB—*Amphora* cf. *pusio*, AC—*Berkeleya* cf. *fennica* AD—*Proschkinia* sp.1. Figure S2. Scanning electron microscopy images of diatom taxa found in samples HPM9 and HPM69 (A) *Trigonium* sp., scale bar 50 μm; (B) *Navicula* sp., scale bar 1 μm; (C) *Amphora* sp., scale bar 2.5 μm; (D) and (E) *Nagumoea* sp., scale bar 1 μm; (F) *Nitzschia* sp., scale bar 5 μm; (G) *Hyalosira* sp., scale bar 2.5 μm.

**Author Contributions:** Conceptualization, R.M. and S.B.; Formal Analysis, L.K.; Funding Acquisition, S.B.; Investigation, L.K. and B.V.d.V.; Methodology, L.K., R.G. and S.B.; Project Administration, S.B.; Resources, R.G. and B.L.; Supervision, S.B.; Validation, R.M. and B.V.d.V.; Writing—Original Draft Preparation, L.K., R.M. and S.B.; Writing—Review and Editing, R.M., B.V.d.V., R.G., B.L. and S.B. All authors have read and agreed to the published version of the manuscript.

**Funding:** This work has been supported by the Croatian Science Foundation under the project UIP-2017-05-5635 (TurtleBIOME).

**Acknowledgments:** The authors are thankful to Nikola Medić for his help in sampling procedures and to three anonymous reviewers for their helpful suggestions and comments.

**Conflicts of Interest:** The authors declare no conflict of interest. The funders had no role in the design of the study; in the collection, analyses, or interpretation of data; in the writing of the manuscript, or in the decision to publish the results.

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
