# Peer review of "Diatom Diversity on the Skin of Frozen Historic Loggerhead Sea Turtle Specimens"

_diversity, doi:10.3390/d12100383_

Round 1
Reviewer 1 Report
It is always interesting to see what microbes are covering large vertebrate animals, and there is growing interest in documenting the diatom flora of aquatic animals, which represent a very specialized niche. This paper documents the diatom flora of dead sea turtles. It's a descriptive paper, so there is not much to criticize. I would, however, strongly recommend modifying Figure 3, which appears to be the raw output of some program. It is largely unreadable -- are those pie graphs? Is that the species or turtle NMDS plot? This really does need to be fixed. I suggest showing the species and community ordinations side by side.
A well written manuscript.
Reviewer 2 Report
Throughout this paper entitled "Diatom diversity on the skin of frozen historic loggerhead sea turtle specimens." Kanjer et al. aimed at characterizing the epizoic diatoms growing in sea turtles. The paper is readable and it contains valuable information to advance in the field of epizoic assemblages
My main criticisms are relatively minor and are linked to methodological aspects. Particularly, I'd like to have more information about the number of samples analyzed per turtle. The reader will acknowledge such information, as if there is only one sample per turtle, there isn't enough data to make a 'valid' Permanova. Actually, no matter the sample size, I strongly recommend the authors to not include this analysis, since it is not adding extra information. If the authors decided to keep it, please write the respective statistics in the results section (p-value, F, number of permutations and degrees of freedom).
Finally, I'd suggest the authors to expand a bit the discussion section by discussing the differences between skin and carapace assemblages. Maybe, you can add a table comparing the most abundant species in skin vs carapace. Also, I'd like to read in the discussion some kind of comparison against freshwater turtles. Do you expect the same patterns in the skin of these kind of turtles? I mean, do you expect diatom assemblages growing over the skin and carapace of freshwater turtles being as similar as in marine turtles? Discussing such topic will enhance the soundness of the study.
Some useful references for the comparison:
- http://dx.doi.org/10.15517/rbt.v66i4.31396
- 10.1371/journal.pone.0171910
- 10.1127/1438-9134/2017/023
Reviewer 3 Report
The manuscript submitted by Kanjer and collegues provides very interesting, new results about the epizoic diatoms associated to marine vertebrates. The manuscript is very well-written. The aims of the investigations and the methods applied are clear. The quality of LM and SEM pictures is excellent. Therefore, the manuscript surely deserves to be published on Diversity with moderate to minor revisions.
I have just a question and a suggestion, both aimed at improve the paper:
1 - At lines 248-249 the authors state: "(…) skin diatom communities have the potential to serve as highly accurate indices of the sea turtle and ecosystem condition." My question is: what is the relevance of post-mortem investigations of epizoic diatoms found on marine vertebrate tissues, in this regard? You do not know exactly when the turtles died, and what may have happened between the death, the transport and the gathering on the beaches. Do you completely exclude a post-mortem colonization of turtle skin? If so, why? On the other hand, if you admit a post-mortem colonization (let's say a 'secondary' attachment), what is the relevance of this sort of study for the assessment of (living) sea turtle and ecosystem condition?
2 - In the light of the author's conclusive considerations about the absence of diatoms from some skin samples and the possible alteration of the original biofilm, I would have greatly appreciated a reflected light and SEM documentation of selected, untreated samples of sea turtle skin. This may have showed the relationships between the substrate and the epizoic diatoms identified.
Finally, below are reported some minor, formal corrections:
Line 67: please delete the comma after 'followed by'
Line 87: 'Caretta caretta' instead of 'Carreta carreta'
